# Brief segments of neurophysiological activity enable individual differentiation

Jason da Silva Castanheira [1,3], Hector Domingo Orozco Perez[2,3], Bratislav Misic [1✉] & Sylvain Baillet [1✉]

Large, openly available datasets and current analytic tools promise the emergence of population neuroscience. The considerable diversity in personality traits and behaviour between individuals is reflected in the statistical variability of neural data collected in such repositories. Recent studies with functional magnetic resonance imaging (fMRI) have concluded that patterns of resting-state functional connectivity can both successfully distinguish individual participants within a cohort and predict some individual traits, yielding the notion of an individual's neural fingerprint. Here, we aim to clarify the neurophysiological foundations of individual differentiation from features of the rich and complex dynamics of resting-state brain activity using magnetoencephalography (MEG) in 158 participants. We show that akin to fMRI approaches, neurophysiological functional connectomes enable the differentiation of individuals, with rates similar to those seen with fMRI. We also show that individual differentiation is equally successful from simpler measures of the spatial distribution of neurophysiological spectral signal power. Our data further indicate that differentiation can be achieved from brain recordings as short as 30 seconds, and that it is robust over time: the neural fingerprint is present in recordings performed weeks after their baseline reference data was collected. This work, thus, extends the notion of a neural or brain fingerprint to fast and large-scale resting-state electrophysiological dynamics.

[1] Montreal Neurological Institute, McGill University, Montreal, QC, Canada. [2] Department of Psychology, Neuroscience and Behavior, McMaster University, Hamilton, ON, Canada. [3] These authors contributed equally: Jason da Silva Castanheira, Hector Domingo Orozco Perez. ✉email: bratislav.misic@mcgill.ca; sylvain.baillet@mcgill.ca

Understanding the biological nature of individual traits and behavior is an overarching objective of neuroscience research[1–4]. The increasing availability of large, openly available datasets and advanced computational tools propels the field toward this aim[5–7]. Yet, with bigger and deeper data volumes, neuroscientists are confronted with a paradox: while big-data neuroscience approaches the realm of population neuroscience, we remain challenged by understanding how inter-individual data variability echoes the singularity of the self[1,3,8,9].

This epistemological question has become particularly vivid with recent research showing that individuals can be differentiated from a cohort via their respective neural fingerprints derived from structural magnetic resonance imaging (MRI)[10,11], functional MRI (fMRI)[12–16], electroencephalography (EEG)[17–19], or functional near-infrared spectroscopy (fNIRS)[20]. Neural fingerprints are associated with individual differences in intelligence test performance, working memory, and attention[21–24]. Most published work so far is methodologically based on inter-individual similarity measures of functional connectivity—understood as statistical dependencies between ongoing signals across brain regions in task-free awake conditions[25,26]—as defining features of neural fingerprints. Yet, the indirect coupling between hemodynamic and neural brain signaling interrogates the neurophysiological nature of brain fingerprints.

In electrophysiology, ongoing brain dynamics at rest are rich and complex[26] and have long been considered a nuisance, a by-product of neural noise[27–29]. Recent experimental evidence, spurred by systems neuroscience models, indicates that spontaneous brain activity captured using electrophysiological techniques expresses similar resting-state connectomes as fMRI and influences conscious, sensory processes[30–32]. Ongoing neurophysiological activity varies considerably between individuals and across the lifespan. One instance is the inter-individual variability of prominent features of human brain neurophysiological activity, such as the alpha rhythm (8–12 Hz) peak frequency[33,34]. Previous EEG fingerprinting work was restricted to scalp data, and therefore, provided limited neuroanatomical insight[17–19]. Another distinctive aspect of electrophysiology is the contamination of recordings by artifacts of different natures including environment and instrument noise, muscle contractions, eye and head movements, which can be distinctive of individuals and can bias fingerprinting with non-neural signal features. Overall, the unique signature components of fast neurophysiological brain dynamics across individuals remain unchartered.

In this work, we use resting-state recordings of magnetoencephalography MEG[35] from a cohort of participants to identify neurophysiological features of individual differentiation. We both derive measures of functional organization (i.e., functional connectivity) inspired by fMRI neural fingerprinting approaches, and spectral signal markers that are proper to the wider frequency spectrum of brain signaling accessible to neurophysiological data. Our data exemplify that individual differentiation based on connectome features is akin to previous fMRI reports, and further demonstrate that we can equally differentiate individuals with simpler measures of the spatial distribution of neurophysiological spectral signal power. In addition, individual differentiation is achieved with recordings as short as 30 seconds and is robust across recordings preformed weeks after their baseline reference data was collected. Together, our work extends the notion of a neural fingerprint to the fast and large-scale resting-state dynamics of electrophysiology.

## Results

We used MEG data from 158 participants available from the Open MEG Archives OMEGA[6]. Data collected on multiple days were available for a subset of these participants ($N = 47$; mean duration between consecutive sessions: 201.7 days; Fig. 1). The participants were both healthy and patient volunteers (ADHD and chronic pain) spanning in age from 18–73-years old (see Supplemental Information). T1-weighted structural MRI volumes were available from OMEGA for all participants and were used to produce source maps of resting-state brain activity[36]. We derived several neurophysiological signal features from MEG brain source time series summarized within the Desikan-Killiany atlas—68 regions of interest (ROIs) parcellating the entire cortical surface[37]. The MEG features comprised power-spectral-density estimates (PSD) within each of the 68 ROIs[37], and $68 \times 68$ functional connectomes (FC) between these ROIs. The approach is illustrated in Fig. 1 and the FC and PSD methodological details are provided in "Methods".

Participant differentiation was performed across pairs of MEG data segments taken from either the same (within-session differentiation) or a repeated session (between-session differentiation) using two distinct datasets (Fig. 1a) and based either on FC or PSD features (referred as connectome and spectral fingerprinting, respectively). The within-session challenge with longer data segments was considered to assess the baseline performances of the MEG fingerprinting approaches proposed. The more challenging situations developed in the present report concern individual differentiation from shorter 30 s time segments within or between recording sessions. For each pair of participants, the Pearson's correlation coefficient between their respective features (i.e., FC or PSD) was the corresponding entry in the group correlation matrix (see Supplemental Information). The fingerprinting procedure for each individual proceeded via a lookup operation through the corresponding row of the correlation matrix; the index of the column featuring the largest correlation coefficient determined the predicted (anonymous) identity of the individual in the cohort. Thus, if a given individual's data features from the first dataset were most correlated to the data features from their second dataset, the individual would be correctly differentiated. Note that taking the maximum along the rows or columns simply switches which dataset is used for deriving the differentiation features (e.g., differentiating individuals using dataset 1 from features derived from dataset 2; results for all possible combinations of datasets are reported in Supplemental Information). The overall accuracy of the neural fingerprinting procedure was computed as the proportion of participants correctly differentiated. We ran three types of differentiation challenges: within-session fingerprinting consisted of the differentiation between 158 participants (i.e., the datasets were from same-day recordings split in half); a between-session differentiation challenge for a subset of 47 participants for whom the datasets were from two separate days; and a between-session differentiation using considerably shorter data segments (30 s) (Fig. 1a). We conducted the differentiation challenges using either broadband MEG data or band-limited versions within the typical frequency bands used in neurophysiology. We also derived a differentiability score for every participant, which indicates the saliency of the differentiation of any given individual in the tested cohort (see "Methods").

**Within-session connectome and spectral data differentiate individuals**. Within-session MEG connectome and spectral fingerprinting achieved 94.9% and 96.2% participant differentiation accuracy, respectively (Fig. 2). This outcome was robust to switching datasets (Supplemental Information). While previous work[12] reported that data reduction strategies improved fingerprinting performances, this was not the case with our data. Data reduction strategies only marginally improved individual differentiation, as explained in Supplemental Information.

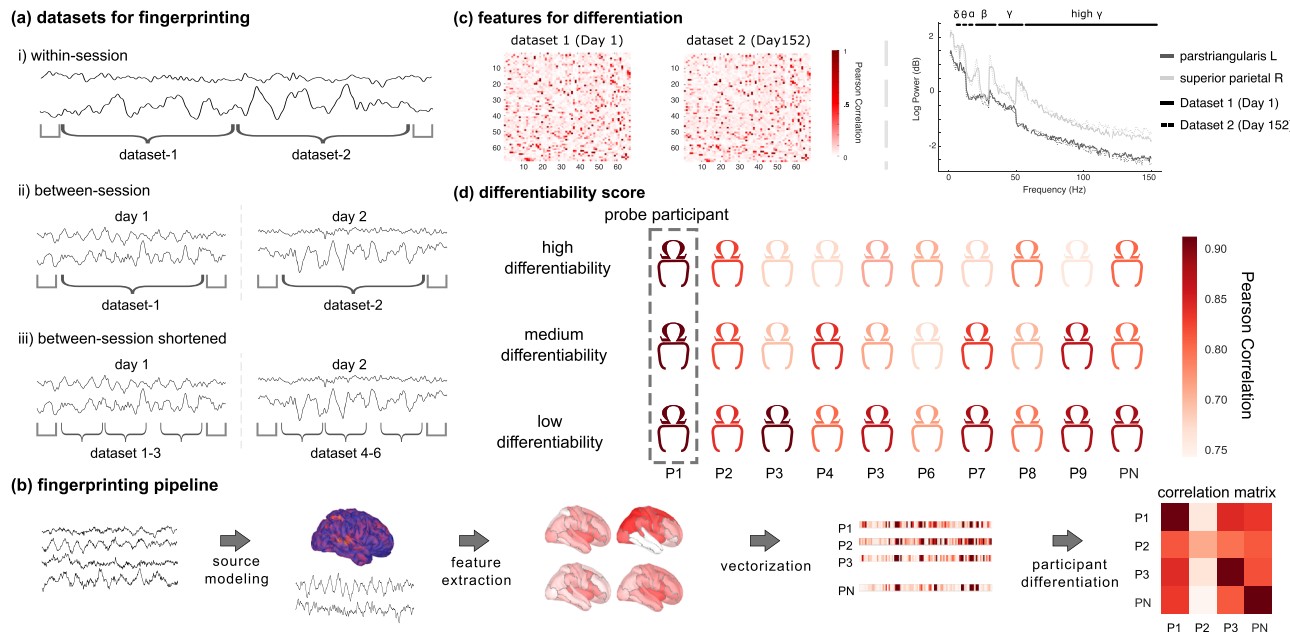

**Fig. 1 Neural fingerprinting analysis pipeline and definition of differentiability. a** Schematic of exemplar MEG data divided into datasets used in each of the specified differentiation challenges. (i) Within-session challenge: the session data was split in half to generate segments of equal duration; (ii) Between-sessions challenge: differentiation was performed using data recorded on two separate days; (iii) Between-session shortened challenge: data recorded on two different days were split into three 30 s segments. **b** Schematic of the data analysis pipeline: source modeling was first performed before extracting features from each region of the Desikan-Killiany atlas[37]. These features were vectorized and subsequently used to fingerprint individuals, yielding a participant correlation matrix. **c** Features for the between-session challenge from an exemplar subject. Left panel depicts amplitude envelope correlation (AEC) functional connectivity matrices across two datasets; both matrices feature the Pearson correlation coefficients between all 68 regions of the Desikan-Killiany atlas[37]. Right panel plots the power spectrum density estimates from two regions of the atlas, across two datasets. **d** Differentiability was derived for each participant as the z-score of their correlation to themselves, relative to the correlation between themselves and the rest of the cohort. A participant with a high correlation to themselves and low correlations to others was qualified as highly differentiable. An individual highly correlated to both themselves and many others in the cohort was qualified as less differentiable.

We also ran the differentiation procedure for each of the typical frequency bands of electrophysiology to understand whether the expression of certain ranges of brain rhythms would be more specific of individual differentiation. We bandpass filtered MEG signals in the delta (1–4 Hz), theta (4–8 Hz), alpha (8–13 Hz), beta (13–30 Hz), gamma (30–50 Hz), and high gamma (50–150 Hz) frequency bands before running the same within-session fingerprinting procedure using the resulting narrowband signals. Narrowband connectome fingerprinting yielded differentiation accuracy scores of 98.7% for delta, 100% for theta, 99.4% for alpha, 100% for beta, 98.7% for gamma, and 94.9% for high gamma. Narrowband spectral fingerprinting produced differentiation accuracies of 94.9% for delta, 95.6% for theta, 95.6% for alpha, 96.2% for beta, 96.2% for gamma, and 97.5% for high gamma. These results are summarized Fig. 2a.

**MEG fingerprinting is robust against physiological, artefactual, and demographics confounds.** We investigated the robustness of these results against variables of no interest and possible confounds. We first processed each individual session's empty-room recordings in an identical fashion to participants brain data. We produced pseudo brain maps of empty-room sensor data using the same imaging kernels as those used for each session's participant brain data. The implication is that imaging kernels designs are based on information that are specific of each participant, such as their respective head positions in the MEG sensor array and individual anatomy brain features that constrain MEG source maps. We, therefore, tested whether such individual information unrelated to brain activity contributed substantially to individual differentiation from MEG source maps. We found that differentiation performances

were considerably reduced using empty-room data (<20% across all tested models; Fig. 2). These results based on source maps were corroborated by the low fingerprinting performances obtained by using empty-room sensor data only (<5% across all tested models; Supplemental Information).

We then performed Pearson correlation analyses between differentiability scores and recording parameters, typical MEG artifacts, and demographic variables. There was no association between the duration of scans and differentiability for connectome ($r = -0.02$, $p = 0.75$) and spectral ($r = 0.02$, $p = 0.8$) fingerprinting (Supplemental Information). Further, none of the tested MEG artifacts due to eye movements, heartbeats, and head motion were related to individual differentiation from either connectome or spectral fingerprinting. Indeed, differentiability was not correlated to motion (connectome: $r = 0.06$, $p = 0.5$; spectral: $r = -0.01$, $p = 0.9$), cardiac (connectome: $r = 0.05$, $p = 0.6$; spectral: $r = 0.07$, $p = 0.4$), or ocular (connectome: $r = -0.09$, $p = 0.3$; spectral: $r = -0.05$, $p = 0.5$) artifacts (Fig. 2b).

Lastly, we further hypothesized that fingerprinting performances may have been skewed by sample heterogeneity in terms of data from healthy vs. patient participants. Yet, there were <1% differences in differentiation accuracy after restricting fingerprinting to healthy participant data (Supplemental Information). We also verified that participant demographics such as age, sex, and handedness did not contribute to differentiability either (Supplemental Information).

**MEG fingerprinting is robust over time.** We tested whether participants who underwent MEG sessions on separate days were differentiable from datasets collected weeks to months apart (with

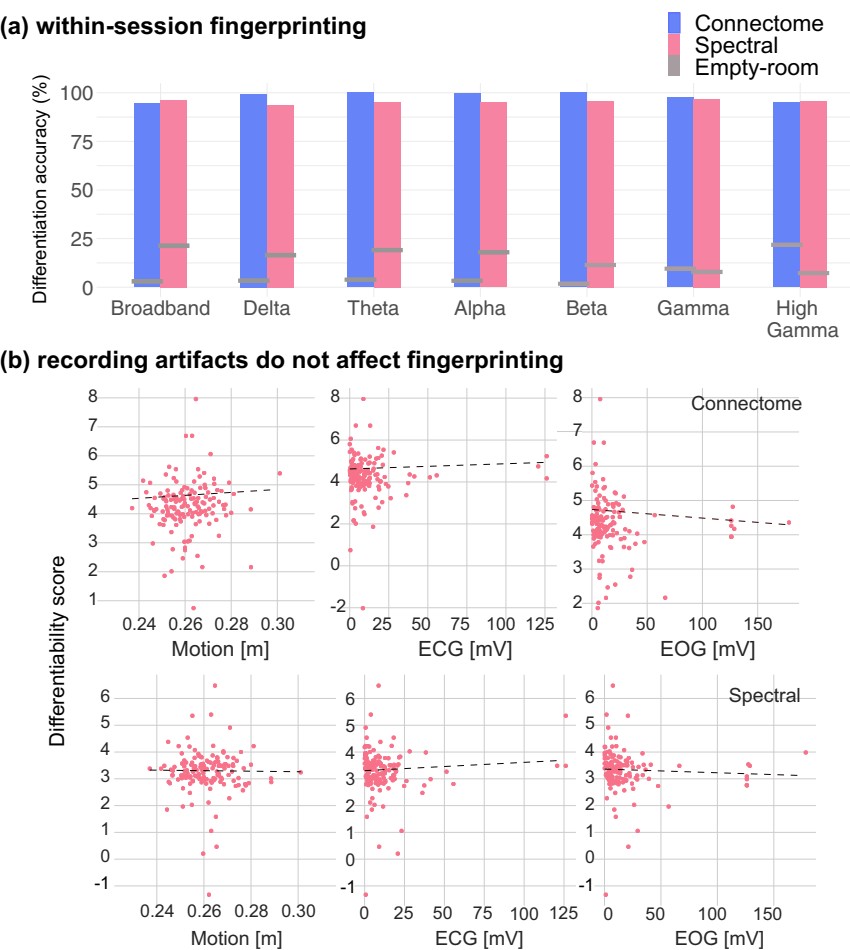

**Fig. 2 Within-session differentiation is not related to recording artifacts. a** Differentiation accuracy of connectome and spectral fingerprinting based on broadband and narrowband brain signals. Horizontal gray bars indicate reference differentiation levels obtained from empty-room data recorded on the same days as participants (see "Methods"). **b** Differentiability scores were not related to typical confounds such as head motion, eye movements, and heartbeats. Top row: using connectome fingerprinting; bottom row: spectral fingerprinting. Source data are provided as a Source Data file.

a range of 1–1029 days apart and an average of 201.7 days, SD = 210.1). We applied the above fingerprinting procedures towards this between-session challenge on the subset of participants concerned (N = 47). Connectome fingerprinting decreased in performance compared to the differentiation accuracy scores obtained from the within-session challenge (89.4%). Performance of connectome fingerprinting from narrowband signals also decreased, with the greatest robustness obtained from using signals in the beta and theta bands (Fig. 3 and Supplemental Information). In contrast, spectral fingerprinting was robust longitudinally, with differentiation accuracy scores of 97.9% (broadband) and >90% (narrowband) that were similar to those obtained in the within-session challenge (Fig. 3 and Supplemental Information). Differentiability scores were not correlated with the number of days between MEG sessions (connectome: $r = 0.09$, $p = 0.5$; spectral: $r = 0.08$, $p = 0.65$).

We further challenged MEG individual differentiation between sessions days apart using shorter data segments. We extracted three 30 s segments from the between-session data on each day (Fig. 1a) and ran the same fingerprinting procedures as above. Differentiation performances from connectome fingerprinting remained high across all 30 s segments tested (Fig. 3c) using broadband MEG signals (differentiation accuracy 83.8%). Performance of spectral fingerprinting was decreased (differentiation accuracy: 61.6% Fig. 3c). We observed similar discrepancies in performance robustness between connectome and spectral

fingerprinting using narrowband signals (Fig. 3), especially in the delta, theta, and alpha bands. We report results obtained from using sensor data only and for the within-session shortened challenge in Supplemental Information.

**Salient neurophysiological features for fingerprinting**. We identified the features which were the most characteristic of individuals for MEG fingerprinting. We derived measures of intraclass correlation (ICC)[12] to quantify how much each feature, such as an edge of the FC connectome or the signal power in a frequency band from an anatomical parcel, contributed to fingerprinting (see "Methods"). This metric was reported in previous brain fingerprinting studies and captures the inter-rater reliability of each participant as their own rater, to identify the neurophysiological signal features that are the most consistent across individuals[12,38]. We performed this analysis for both the broadband connectome and the band-specific spectral fingerprinting within-session challenges. The data show that the dorsal attention and visual networks were the most specific across individuals for connectome fingerprinting, in all frequency bands (Fig. 4). Beta-band connectivity of the limbic network was particularly distinctive of individuals. For spectral fingerprinting, theta, alpha, beta, and gamma bands discriminated individuals along midline, parietal, lateral temporal, and visual areas (Fig. 4b). These results are consistent with our narrowband analysis (see Fig. 2a), which

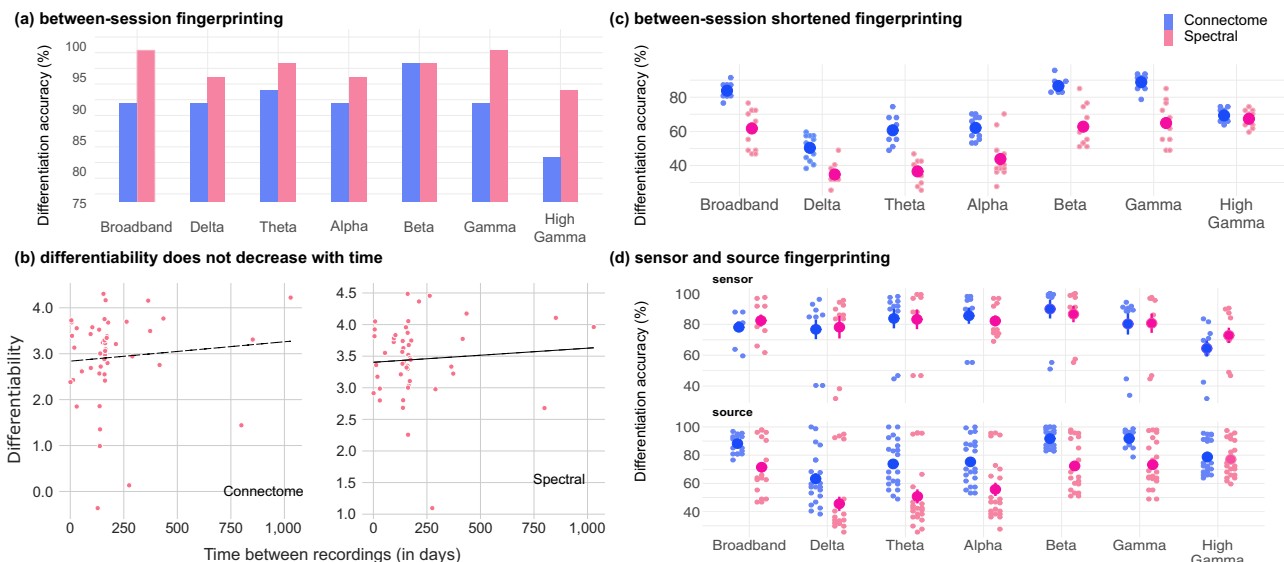

**Fig. 3 Between-session fingerprinting accuracy. a** Differentiation accuracy for connectome and spectral between-session fingerprinting. Fingerprinting performances are similar to those from the within-session challenge. **b** Pearson correlation analyses did not reveal an association between differentiability and the delay between session recordings (connectome fingerprinting: $r = 0.09$, $p = 0.5$; spectral fingerprinting: $r = 0.08$, $p = 0.60$). **c** Between-session-shortened differentiation accuracy using shorter 30 s data segments collected days apart (average: 201.7 days). Each data point represents one combination of datasets used for fingerprinting (see "Methods" for details). **d** Scatter plots of all fingerprinting challenges across frequency bands for source (brain) and sensor (scalp) level fingerprinting (Supplemental Information details the results obtained for all sensor data fingerprinting challenges). Source data are provided as a Source Data file.

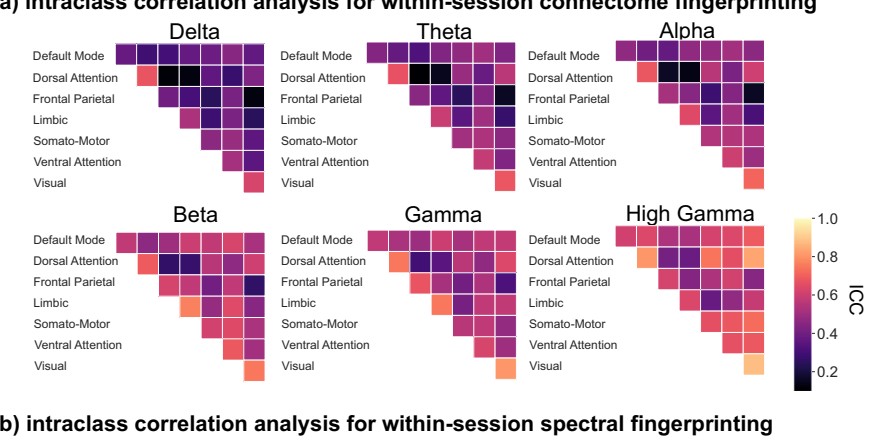

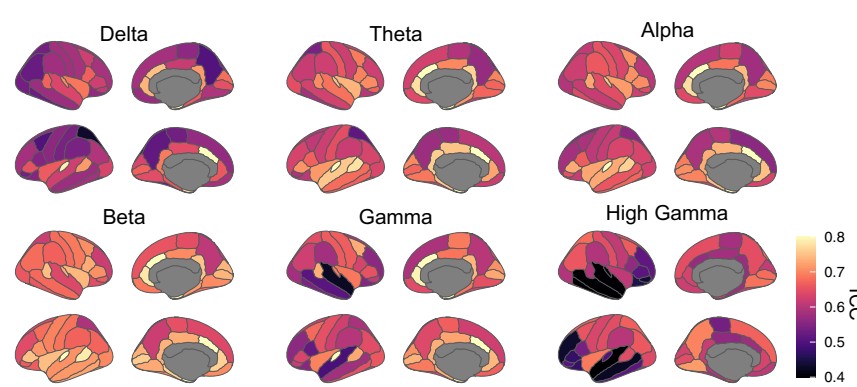

**Fig. 4 Characteristic features of connectome and spectral fingerprinting.** Intraclass correlation (ICC) for connectome and spectral within-session fingerprinting. **a** ICC for connectome fingerprinting plotted for each tested frequency band, using network labels from Yeo et al.[81]. The most prominent networks for connectome fingerprinting were the Visual, Dorsal Attention, and Limbic networks. **b** ICC for spectral fingerprinting plotted for each tested frequency band and mapped using the Desikan-Killiany cortical parcellation[37]. The most salient features were the theta, alpha, and gamma band signals expressed in midline structures and the beta band across the cortex.

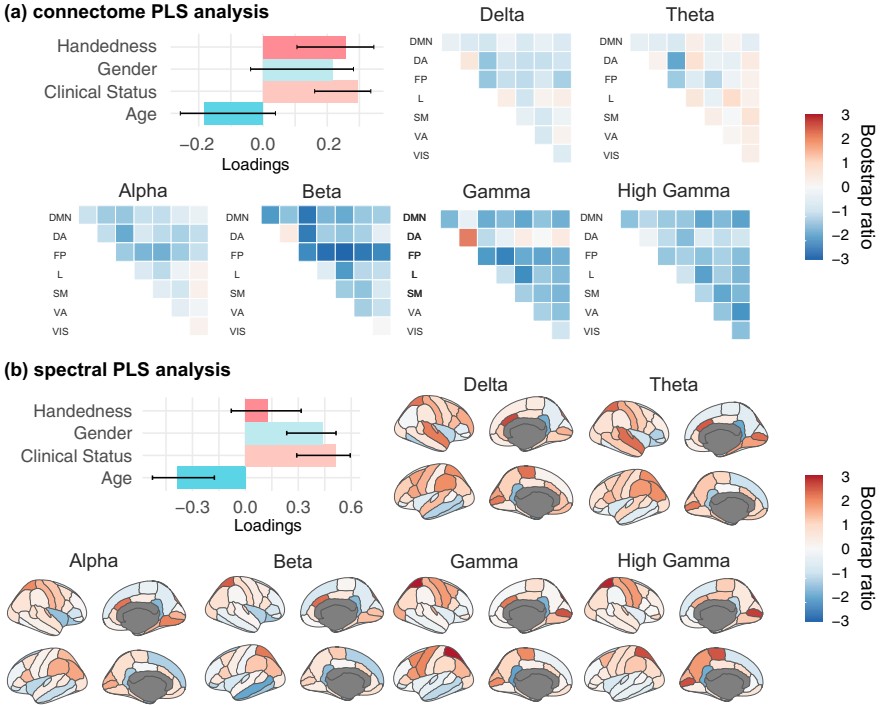

**Fig. 5 Partial Least-Squares analysis relates demographics to connectome and spectral features.** (**a**) and (**b**) from left to right, depicts the design saliency patterns for the first latent variables and their associated neural-data bootstrap ratios. Confidence Intervals (95% CI) were calculated through a bootstrapping procedure ($n = 10,000$), and as such may not necessarily be symmetric. Bootstrap ratios computed for (**a**) connectome and (**b**) spectral features are plotted according to the resting-state networks labeled according to Yeo et al.[81] and the Desikan-Killiany parcellation[37], respectively: Default Mode Network (DMN), dorsal attention (DA), frontal-parietal (FP), limbic (L), somato-motor (SM), ventral attention (VA), and visual (VIS). Source data are provided as a Source Data file.

highlights beta activity as the most informative in differentiating individuals.

**Neurophysiological fingerprinting features are associated with demographics.** Beyond differentiating individuals in a cohort, we tested whether resting-state neurophysiological features could also predict meaningful participant traits, using an exploratory partial-least-squares (PLS) analysis (see "Methods"[39]). Briefly, PLS explains the structure of the covariance between two observation matrices—here a demographic matrix and a neurophysiological signal matrix composed of ROI-specific connectome of spectral measures—with latent components. PLS analysis of our data revealed three significant latent components, which were distinct for connectome and spectral fingerprinting (Supplemental Information). The first latent component in connectome fingerprinting was related to clinical population ($r = 0.2$, 95% CI [0.160, 0.3]) and handedness ($r = 0.2$, 95% CI [0.1, 0.3]). This demographic profile was associated with reduced beta-band functional connectivity over the frontal-parietal network (Fig. 5). For spectral fingerprinting, the first salient latent component was related to a younger age ($r = -0.3$, 95% CI [$-0.1$, $-0.5$]), female ($r = 0.4$, 95% CI [0.2, 0.5]) and clinical population ($r = 0.5$, 95% CI [0.2, 0.5]). This demographic profile was associated with stronger expressions of broadband neurophysiological signal power in superior parietal regions and the pericalcarine gyrus bilaterally, and reduced neurophysiological signals in the isthmus cingulate (Fig. 5).

## Discussion

The recent leveraging of large, open fMRI datasets has brought empirical evidence that individuals may be differentiated within a cohort from their brain imaging functional connectivity, inspiring the metaphor of a neural fingerprint. Unlike hand fingerprints, their cerebral counterpart predicts task performance and a variety of traits[14,21–24]. These intriguing findings require a better understanding of their neurophysiological foundations, which we sought to characterize from direct neural signals captured at a large scale with MEG.

Our data show that individuals can be differentiated in a cohort of 158 unrelated participants from their respective resting-state connectomes and spectral profiles in a range of fast brain signals. MEG fingerprinting was successful using data lengths (30 s) much shorter than those reported for fMRI fingerprinting[14,40]. Brain electrophysiological signals are rich, complex, and convey expressions of large-scale neural dynamics channeled by individual structural anatomy and physiology[41]. Indeed, we also showed that MEG fingerprinting is robust across time, making individuals potentially differentiable from data collected days, months, or years apart. Lastly, we characterized whether individual differences in resting-state neural dynamics are demographically meaningful through an exploratory PLS analysis. We showed that both resting-state connectomes and spectra predict latent demographic components. Recent findings corroborate our results, demonstrating individual differences between functional connectivity derived from resting-state electrophysiology[42]. Future work will be required to replicate and expand these findings in more samples of individuals.

**Connectome and spectral neurophysiological fingerprints.** Our results highlight two sets of brain-wide electrophysiological features that contributed to successful fingerprinting: connectome and spectral measures across the neurophysiological frequency spectrum. Overall, connectome and spectral fingerprinting with MEG performed equivalently to fMRI approaches, achieving

overall differentiation rates above 90%, with robust individual differentiation over time and against noise[12,14,43].

We found that for connectome fingerprinting, the anatomical regions the most characteristic of individuals differed between MEG and fMRI. While fMRI highlighted the default-mode network and the fronto-parietal resting-state networks, MEG connectome fingerprinting emphasized functional connectivity within limbic and visual networks as contributing to individual specific neurophysiological signatures. In contrast, both MEG and fMRI fingerprinting emphasize the importance of the dorsal attention network[14]. These observations are not mutually exclusive, considering the different nature of brain signals captured by the respective modalities. One possible interpretation—requiring further investigation—is that the fast neurophysiological signals that contribute to differentiation with MEG have hemodynamic counterparts that are not as salient in fMRI as the fingerprinting networks reported so far. Nevertheless, our data indicate that neurophysiological signals in the beta band contribute to the highest differentiation accuracy amongst all other typical bands. This finding is compatible with previous work reporting that correlated amplitude changes of MEG brain signals are related to the microstructure of white matter tracts and reveal, with the same amplitude envelope correlation method as used here, MEG resting-state brain networks that align with fMRI's[44,45]. Beta-band activity also emerges from recent literature as a signalling vehicle of re-afferent "top-down" communications in brain circuits[46,47]. One can therefore speculate that beta-band signals would convey electrophysiological representations of internal cognitive models that are by essence intimately specific of each individual[35].

Such brain signal amplitude signatures are further emphasized by the ability of simple spectral brain maps to enable MEG fingerprinting. Within- and between-session spectral fingerprinting were achieved with remarkable accuracy (>90%) with broadband MEG brain signals or restricted to the typical bands of electrophysiology. Spectral differentiation based on signals from the faster bands (gamma and high-gamma) was overall the most robust longitudinally and against using shorter data segments. This observation is consistent with the width of (high) gamma frequency bands spanning broader ranges (here between 30–50 Hz and 50–150 Hz) than slower bands such as delta (1–4 Hz), theta (4–8 Hz) and alpha (8–12 Hz). The spectral estimates averaged across the broader (high) gamma bands were, therefore, the most robust against using shorter data segments. The reduced number of sliding time windows available over shorter data durations increased the variance of the summary statistics extracted to derive the spectral fingerprints from the signals defined over narrower bands. The higher frequency bands were less affected because the larger number of frequency bins involved in the extraction of their summary power statistics tended to compensate the higher empirical variance of spectral estimates from a lesser number of observations over time. Connectome fingerprinting was more immune against using shorter data durations. The underlying approach indeed did not require spectral transformations but resorted to a bank of narrowband filters applied over the original duration of MEG recordings before the resulting filtered signals were segmented in shorter epochs for the fingerprinting challenges. The consequence is that the number of data points used for all narrowband signals was identical across all frequency bands, yielding moderate variability in differentiation performances compared to those obtained with the spectral approach. Another point of robustness for connectome fingerprinting is that connectivity weights between network nodes may fluctuate very slowly over time in task-free brain activity: Florin and Baillet[30] reported fluctuation rates of 0.01 Hz in MEG, indicating typical time cycles of 100 s—a duration substantially longer than the 30 s shortest time window used here. Over longer periods of time though, such as in the between-session challenge, spectral fingerprinting outperformed its connectome counterpart. We note a slight increase of spectral differentiation accuracy in the between-session challenge (e.g., +1.6% for broadband fingerprinting) compared to within-session, which was a statistical fluctuation due to using a smaller sample of participants.

On average across all source fingerprinting challenges reported herein, and despite successful fingerprinting across lower frequency bands (delta 54.4%, theta 62,3%, alpha 65.5%), performances were markedly better using high-frequency signal components (beta 82.0%; gamma 82.5%; high gamma 77.9%). Gamma and faster activity have long been associated with concurrent and colocalized hemodynamic fluctuations[48,49]. Because they may be seen as dual manifestations of BOLD signaling used in fMRI fingerprinting, this may explain why these signals contributed robustly to MEG brain fingerprinting in our data. However, gamma-band and faster brain signals are on average weaker in amplitude and therefore may be masked by contamination from artifacts and noise[50–52]. The preprocessing applied to our data attenuated such nuisance to a point where individuals were not differentiable from typical sources of signal contamination such as individual head motion behavior.

Although a rhythm of prominent amplitude in humans during rest, alpha-band activity (8–12 Hz) was not particularly specific to differentiate individuals in the cohort. In that respect, our data is aligned with previous MEG works on resting-state connectomes extracted from neurophysiological MEG signals, which did not report on a salient role of alpha activity in driving inter-regional connectivity[30,44]. We argue that the spatial topography of alpha resting activity may be relatively stereotypical across individuals, involving thalamo-cortical loops that project focally to the parieto-occipital junction, with limited variability across individuals[6]. In task, alpha activity has been related to attention orienting, alertness and anticipation, and the registration of (multimodal) sensory information, thereby reflecting transient mental states[32,53–56] rather than individual traits.

The data also indicates that MEG fingerprinting is robust against typical recording artifacts that may be idiosyncratic of individuals and therefore, could have confounded fingerprinting. Session environmental conditions captured by empty-room MEG recordings were not sufficient to differentiate individuals within or between sessions. The participant's anatomical and head-position information embedded in their respective MEG source imaging kernels were also not sufficient to differentiate individuals. Note that head position changed between sessions. Further studies are required to clarify how these results may vary depending on the type of MEG source modeling adopted. We anticipate little influence of the type of source model used though, based on evidence that beamforming kernels are mathematically equivalently to other major classes of linear source estimation kernels, such as weighted minimum-norm estimators[57]. Future work should corroborate these results with regards to fingerprinting. The choice of connectivity measure to derive electrophysiological connectomes may also influence fingerprinting[58]. We look forward to current progress in electrophysiological brain connectomics to put forward measures of network connectivity informed by mechanistic principles and emerging as a standard metrics in the field to confirm and expand present fingerprinting results[59].

While our present data show robust longitudinal fingerprinting performances, future work involving more participants with multiple MEG visits is required to both replicate these observations and investigate whether individual deviations from baseline fingerprints could be early signals of asymptomatic

neuropathophysiology[35]. We hope the remarkable ability to fingerprint individuals from the present electrophysiological features serves as a stepping stone for future investigation, which may include multimodal noninvasive assessments based on MEG, combined with, e.g., fMRI and/or EEG.

**Neural fingerprints of individual traits.** Our data suggest that individual differences in resting-state neurophysiological functional connectivity and spectral power relate to latent demographic clusters. These observations are in line with previous fMRI work that showed that connectomes are predictive of individual differences in attention, working memory, and intelligence test performance. For instance, connectivity patterns between the default mode and the dorsal attention networks predict attentional behavior during the task and self-reported mind wandering[22,60] (see[61] for review). Overall, a possible conceptual framework is that task-free neural dynamics are the signatures of an individual scaffold of brain functions that is predictive of task behavior. This view is also that of the spontaneous trait reactivation hypothesis wherein the organization of the human cortex at rest (manifested e.g., by functional connectivity) is a window into the self's unique traits and abilities[62]. Early evidence indeed suggests that functional connectivity and brain activity are associated with personality traits and even inter-personal closeness in social networks[63,64].

Yet, the mechanistic implementation of these intriguing observations remains elusive. Inter-individual variability in the distribution of synaptic weights across the cerebrum, shaped through lifetime experiences according to Hebbian principles, may account—at least in part—for connectome fingerprinting[62]. The heritability of functional connectivity has also been discussed, especially for a variety of brain networks (e.g., dorsal and ventral attention network and the default mode network)[65–67]. Heritability of brain spectral characteristics is also actively discussed[68–70]. This emerging literature and the empirical evidence of brain fingerprinting certainly motivate more research on new, fascinating questions about the biological nature of the self.

**Sampling population diversity for personalized interventions.** Robust individual signatures of brain activity may be transformative to neurophysiological phenotyping and population neuroscience. With the increasing availability of multi-omic data repositories, there is a research opportunity to span the diversity of statistical normative characteristics of brain fingerprints across the population in relation to behavior, environmental, and clinical variables[1,3,35]. Our study highlights the utility of datasets of individuals who have been scanned on multiple occasions to capture and characterize interindividual variability as meaningful information. Ideally, large databanks of individual variants sampled across multiple dimensions of socio-economic, age, and geographic factors enable normative modeling approaches to establish the risk traits of developing syndromes of, e.g., early cognitive decline, neurodegeneration, or mental illness. Previous work has shown that mental disorders may affect the stability of individual fingerprints over time and therefore points at possible translational applications of the approach[15,71]. We also foresee that changes over time or lack thereof of a person's brain fingerprint may also constitute a new class of non-invasive markers of responses to neurological and other treatment in a variety of chronic, neurodegenerative, or acute (e.g., stroke) conditions. Brain fingerprints derived from short, task-free sessions may play a leading role to realize this vision in practice.

Brain fingerprinting may also contribute to future endeavors in establishing how oscillatory dynamics at rest support cognitive functions across the lifespan. MEG brain fingerprinting presents several potential advantages in terms of safety, shorter scan duration, and the immediate proximity of a care person during data collection, especially for special populations.

The methodological approaches proposed herein can, in principle, transfer to EEG fingerprinting[17–19], which would be more readily available in clinics. Whether results would be as robust with EEG as MEG remains to be demonstrated. Indeed, EEG source mapping is more prone to contamination from muscle artifacts and is more sensitive to approximations in the biophysical modeling of head tissues, which may compromise further fingerprinting capabilities[35].

In sum, our study extends the concept of neural or brain fingerprint to fast and large-scale resting-state electrophysiological dynamics, which encapsulate meaningful individual differences in both functional connectivity and neuroanatomical maps of power spectrum characteristics. We are hopeful that the present contribution paves the way to replication and extension using larger open datasets. Many fascinating outstanding questions remain about the biological nature of inter-individual variability expressed via neural oscillations and brain network dynamics, and more specifically how these differences associate with behavior and diseases natural history. The research ahead is for future population neuroscience studies.

## Methods

**The Open MEG Archives (OMEGA).** We used data from the Open MEG Archives (OMEGA[6];) consisting of resting-state MEG recordings acquired using the same MEG system (275 channels whole-head CTF; Port Coquitlam, British Columbia, Canada). The sampling rate was 2400 Hz, with an antialiasing filter applied at 600 Hz cut-off, and built-in third-order spatial gradient noise cancellation (see ref. [6] for details on data acquisition).

We analyzed MEG resting-state data from 158 unrelated OMEG participants (77 females, $31.9 \pm 14.7$ years old). Recordings were ~5-min long. Supplementary Table 1 provides details on scanning procedures and Supplementary Table 2 on demographics. A subset of these individuals ($N = 47$) had recordings over multiple visits (different days) and were used in the between-session fingerprinting challenge. The OMEGA data management protocol was approved by the research ethics board of the Montreal Neurological Institute. We followed the ethical procedure of our local ethics board (the Montreal Neurological Institute).

**MEG data preprocessing and feature extraction.** MEG data were preprocessed using Brainstorm[72]; version Oct-12-2018 in MATLAB 2017b (Mathworks, Inc., Massachusetts, USA) following good-practice guidelines[73]. Unless specified, all steps below were performed using the Brainstorm toolkit, with default parameters. Line noise artifact (60 Hz) along with 10 of its harmonics were removed using a notch filter bank. Slow-wave and DC-offset artifacts were removed using a high-pass FIR filter with a 0.3-Hz cut-off. We derived Signal-Space Projections (SSPs) to remove cardiac and ocular artifacts. We used electro-cardiogram and -oculogram recordings to define signal projectors around identified artifact occurrences. We also applied SSPs to attenuate low-frequency (1–7 Hz) and high-frequency noisy components (40–400 Hz) due to saccades and muscle activity, respectively. Bandpass filtered duplicates of the cleaned data were produced for each frequency band of interest (delta: 1–4 Hz, theta: 4–8 Hz, alpha: 8–13 Hz, beta: 13–30 Hz, gamma: 30–50 Hz, and high gamma: 50–150 Hz). Distinct brain source models were then derived for all narrowband versions of the MEG sensor data.

Each individual T1-weighted MRI data were automatically segmented and labeled with Freesurfer[74]. Coregistration with MEG sensor locations was derived using dozens of digitized head points collected at each MEG session. We produced MEG forward head models for each participant using the overlapping spheres approach, and cortical source models with linearly-constrained minimum-variance (LCMV) beamforming, all using Brainstorm with default parameters (2016 version for source estimation processes). We performed data covariance regularization. To reduce the effect of variable source depth, the estimated source variance was normalized by the noise covariance matrix. Elementary MEG source orientations were constrained normal to the surface at 15,000 locations of the cortex. Noise statistics for source modeling were estimated from two-minute empty-room recordings collected as close as possible in time to each participant's MEG session. Source timeseries were clustered into 68 cortical ROIs defined from the Desikan-Killiany atlas[37] and dimension-reduced via the first principal component of all signals within each ROI.

Connectome and spectral fingerprinting features were computed from ROI source timeseries. Individual connectomes were derived in all frequency bands from the amplitude envelope correlation (AEC) approach[75]. ROI timeseries were

Hilbert transformed and all possible pairs of resulting amplitude envelopes were used to derive the corresponding Pearson correlation coefficients, yielding a $68 \times 68$ symmetric connectome array. We used Welch's method to derive power spectrum density (PSD) estimates for each ROI[76], using time windows of 2 s with 50% overlap sled over all ROI timeseries and averaged across all PSDs within each ROI. The resulting frequency range of PSDs was 0–150 Hz, with a frequency resolution of 0.5 Hz. The connectome and spectral features were then exported to Python (3.7.6) for subsequent fingerprinting analyses.

**Fingerprinting and differentiability**. We used a fingerprinting approach directly adapted from fMRI connectome fingerprinting methods[12,14], which relies on correlational scoring of individuals between datasets. A given probe participant is differentiated from a cohort by computing all Pearson correlation coefficients between the spectral or connectome features of said probe at one timepoint (e.g., dataset 1) and the entire cohort at a different timepoint (e.g., dataset 2). The entry presenting the highest correlation to the probe determined the probe's estimated identity, i.e., identified entry in the cohort. This approach was applied between all pairs of participants in the cohort, yielding an asymmetric correlation matrix spanning the cohort. We report scores of differentiation accuracy as the ratio between the number individuals correctly differentiated with the described procedure and the total number of individuals in the cohort. Differentiation accuracy scores are obtained from fingerprinting challenges from dataset 1 to dataset 2 and vice-versa, within- and between-sessions. Figure 1 details the definition of the dataset labels used, and Supplemental Information contains the results from across all combinations of datasets/sessions.

Amico and Goñi[12] proposed an identifiability score to quantify, for a given participant, the reliability of its differentiation from others in the cohort. Here, we extend this notion with the introduction of a differentiability measure, $D_{self}$. Let $\mathbf{A}$ be the correlation matrix spanning the cohort (square, asymmetric) between dataset 1 and dataset 2, and N be the number of participants to differentiate. We define $D_{self}$ as the z-score of participant $P_i$'s correlation to themselves between dataset 1 and dataset 2, with respect to $P_i$'s correlation to all other individuals in the cohort, noted: $D_{self\ (i)} = (Corr_{ii} - \mu_{ij})/\sigma_{ij}$, where $Corr_{ii}$ is the $P_i$'s correlation between dataset 1 and dataset 2, $\mu_{ij}$ is the mean correlation between participant $\mathbf{P_i}$ in dataset 1 and all other individuals in dataset 2 (i.e., the mean along the $i^{th}$ row of matrix A), and $\sigma_i$ is the empirical standard deviation of inter-individual features correlations. Thus, if a participant is easily differentiable, its differentiability increases; whereas small differentiability scores indicate a participant that is particularly difficult to differentiate from the rest of the cohort.

**Recording artifacts and differentiability**. To investigate the effects of recording parameters and artifacts on fingerprinting, we related each individual's differentiability to several possible confounds. The duration of each scan was compared to differentiability to verify that longer recordings available from a subset of individuals did not make them easier to differentiate. We also correlated the root mean square (RMS) of signals that measured ocular, cardiac, and head movement artifacts over the duration of the entire recording to participants' differentiability score. For cardiac artifacts for instance, we derived the RMS of ECG recordings; for ocular artifacts we used the HEOG and VEOG electrode recordings; and for motion artifacts we extracted the RMS of all three head coil signals that measured 3-D head movements during MEG recordings. These derivations were conducted for both the connectome and spectral broadband within-session fingerprinting challenge.

**Fingerprinting across frequency bands**. We replicated the above fingerprinting approach using data restricted to each frequency band of interest (delta 1–4 Hz, theta 4–8 Hz, alpha 8–13 Hz, beta 13–30 Hz, gamma 30–50 Hz, and high gamma 50–150 Hz). We report the differentiation accuracy obtained from each narrow-band signal in both the spectral and connectome fingerprinting challenges in Figs. 2 and 3, for the within- and between-session fingerprinting challenges respectively.

We also performed fingerprinting tests based on sensor data only. We used the same connectome and spectral approaches as the MEG source maps, considering the time series of each of the 275 MEG channels instead of the 68 ROI time series derived from the brain map parcels. We report the differentiation performances from both the sensor and source analyses in Fig. 3 and in Supplemental Information.

**Between-session and shortened fingerprinting challenges**. We verified the robustness of MEG fingerprinting with respect to (1) the ability to differentiate participants over time and (2) reduced data durations. We subdivided participants into three additional challenges: the within-session-shortened, between-session, and between-session-shortened challenge. First, we used the participant data described in the within-session analysis and extracted connectome and spectral fingerprinting features over three 30-second non-overlapping time segments. This duration was based on the length of the shortest recording in the data sample (Fig. 1aii). We applied the same fingerprinting procedure as described in Fingerprinting and Differentiability across all possible combinations of the three 30 s

datasets. Second, we assessed the stability of the fingerprinting outcomes using a subset of participants with consecutive MEG sessions separated by several days ($N = 47$; separated on average by 201.7 days, see Supplemental Information for details). Again, we applied the same fingerprinting procedure as described in Fingerprinting and Differentiability for this between-session challenge. Lastly, we applied the same shortened analysis—described above—to the subset of individuals with multiple scans (i.e., the between-sessions data). We report all possible combinations of datasets (i.e., three 30 s segments from day 1 and three 30 s segments from day 2; see Fig. 1a for example) in Fig. 3.

**Empty-room fingerprinting**. We tested whether environment and instrument noise daily conditions would bias individual differentiation using empty-room recordings collected from each MEG session. The empty-room data was processed identically to the participants data, using the same individual imaging kernels, and were used to differentiate participants. We ran all possible combinations of empty-room vs. participants datasets (e.g., empty-room 1 vs. participant dataset 1, empty-room 2 vs. participant dataset 1, etc.) and computed the sample mean of the differentiation accuracies across all dataset combinations. The differentiation accuracies obtained represent estimates of baseline reference performances that can be compared to each form of fingerprinting based on actual participant data (i.e., connectome or spectral, broadband or band-specific; see Fig. 2 and Supplemental Information). In a similar fashion, we also used sensor-level empty-room recordings of each participant for fingerprinting—attempting to differentiate individuals' recordings from their empty-room features. The results of this analysis are reported in the Supplemental Information.

**Most characteristic features for fingerprinting**. We quantified the contribution of each feature (i.e., edges in the connectivity matrix or a frequency band in an anatomical parcel) towards differentiating individuals using intraclass correlations (ICC). ICC is commonly used to measure the agreement between two observers (e.g., ratings vs. scores). The stronger the agreement, the higher the ICC[12,38]. ICC derives a random effects model whereby each item is rated by different raters from a pool of potential raters. We selected this measure to capture the inter-rater reliability of each participant as their own rater to identify which edges (e.g., connections in FC) are the most consistent (i.e., which features of a participant $_{Pi}$ in dataset 1 are most like dataset 2). Here, the higher the ICC, the more consistent a given feature was within individuals. In addition, we computed two other measures of edgewise contribution proposed by Finn and colleagues[14]: group consistency and differential power (Supplemental Information). We applied all measures (i.e., ICC, group consistency, and differential power) in the context of the broadband within-session fingerprinting challenge. The source maps shown in Figs. 4, 5 and Supplemental Information were generated using R (V 3.6.3[77]; with the ggseg package[78]).

**Partial Least-Squares: MEG features of participant demographics**. We conducted a Partial Least-Squares (PLS) analysis with the Rotman-Baycrest PLS toolbox[79] in MATLAB 2017b (Mathworks, Inc., Massachusetts, USA). PLS is a multivariate statistical method that relates two matrices of variables (e.g., neural activity and participant demographics) by estimating a weighted linear combination of variables from both data matrices to maximize their covariance. The associated weights can be interpreted neural patterns (e.g., functional connections) and their associated demographic profiles. PLS used singular value decompositions of the z-scored neural activity-demographics covariance matrix. This decomposition yielded orthogonal latent variables (LV) associated to a pattern of neural activity (i.e., functional connectivity or spectral power) and demographics. To assess the significance of these multivariate patterns, we computed permutation tests (10,000 permutations). Each permutation shuffled the order of the observations (i.e., the rows) of the demographic data matrix before running PLS on the resulting surrogate data under the null hypothesis that there was no relationship between the demographic and neural data. A p-value for the LVs was computed as the proportion of times the permuted singular values exceeded that of the original data. We explored the first significant LV from the broadband connectome and spectral fingerprinting features. We also assessed the contribution of each variable in the demographics and neural activity matrices by bootstrapping observations with replacement (10,000 bootstraps). We computed 95% confidence intervals for the demographic weights and bootstrap ratios for the neural weights. The bootstrap ratio was computed as the ratio between each variable's weight and the bootstrap-estimated standard error.

**Reporting summary**. Further information on research design is available in the Nature Research Reporting Summary linked to this article.

## Data availability

Resting-state data were obtained from the OMEGA repository[6]. Raw MEG recordings can be accessed by requesting the data (https://www.mcgill.ca/bic/omega-registration). The power spectra and connectomes derived from the preprocessed OMEGA samples and used to differentiate individuals in the present study can be reproduced using the code that has been made available and are available from the corresponding author upon reasonable request. Source data are provided with this paper.

## Code availability

All codes for preprocessing and data analysis can be found on the project's GitHub (https://github.com/neurohazardous/megFingerprinting and at Zenodo (https://zenodo.org/record/5181836)[80].

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

## Acknowledgements

S.B. is grateful for the support received from the NIH (R01 EB026299), a Discovery Grant from the Natural Science and Engineering Research Council of Canada (436355-13), the CIHR Canada Research Chair in Neural Dynamics of Brain Systems, the Brain Canada Foundation with support from Health Canada, and the Innovative Ideas program from the Canada First Research Excellence Fund, awarded to McGill University for the Healthy Brains for Healthy Lives initiative. This research was undertaken thanks in part to funding from the Canada First Research Excellence Fund, awarded to McGill University for the Healthy Brains for Healthy Lives initiative. BM acknowledges support from the Natural Sciences and Engineering Research Council of Canada (NSERC Discovery Grant RGPIN \#017-04265) and from the Canada Research Chairs Program. J.D.S.C. acknowledges the support of the Alexander Graham-Bell Doctoral NSERC fellowship.

## Author contributions

All authors conceptualized the study, J.d.S.C. and H.D.O.P. performed the analyses, S.B. and B.M. provided guidance with methods and data interpretation, J.d.S.C. wrote the first draft of the manuscript, all authors contributed to the writing and editing of the manuscript.

## Competing interests

The authors declare no competing interests.
