## [Peer Review File · Nature Communications]

Brief segments of neurophysiological activity enable individual differentiationREVIEWER COMMENTS

Reviewer #1 (Remarks to the Author):

Authors use resting-state MEG data from an open repository to show that identification of individuals based on their functional connectome or spectral signature is possible. This identification is performed on within-session data and between-session data. In addition, authors relate these fingerprints to demographic data. Overall, the research question is clearly described and relevant, the methodological approach is mostly appropriate and the results are largely convincing. Still, a number of concerns exist and are described in detail below.

Novelty is one concern. There is a sizeable literature of EEG-based biometric studies (even beyond refs 17,18). Some of them are based on resting-state data, use similar (or even more sophisticated) approaches and show similar identification accuracy. I am happy to acknowledge that the current study uses a large cohort and source-localised data but I suggest to discuss this study more explicitly in the context of the EEG literature. Authors state on page 12: 'The prospect of transferring MEG fingerprinting methodology to EEG as a more accessible sister technology is also appealing.' This is clearly incorrect since - if anything - the transfer happened in the opposite direction. Please emphasise the novel aspects of this study and acknowledge the existing EEG literature (particularly in the discussion).

Within-session versus between-session:

In general, I am not convinced by the within-session results.

First, afaik the standard of this type of work in other modalities (e.g. fMRI) is always based on separate recordings. I don't see a good reason to change this here.

Second, it is difficult to ensure that within-session results are not caused or contaminated by effects that are independent of the individual. Imagine some external noise source on the day of measurement that is picked up by the MEG system and introduces a spectral structure (or connectivity structure) that leads to high correlation for the within-session fingerprinting. In fact, figure 2c shows that performance degrades with increasing temporal distance between the short data blocks. This suggests that some slow-changing features exist in the data that apparently boost within-session correlation. I currently don't see how a strong case for fingerprinting can be made based on within-session correlations.

In summary, I suggest to remove the within-session results or at least move them to Suppl Mat. If authors disagree then I would expect to see a good argument for this approach (which is apparently not the standard (e.g. in fMRI)).

Related to the previous point: If authors want to make the claim that 30s are sufficient for fingerprinting then this should be shown for between-session data as well.

All results are based on the correlation matrix of feature vectors within and between participants. Please show an example matrix in Suppl Mat.

Please add methodological details: What type of beamformer was used? Any regularisation? How was the empty-room recording used? For the Welch PSD provide frequency resolution and frequency range.

It would be very helpful to estimate the expected accuracy in the absence of any real correlations between data1 and data2 due to subject-specific brain activity. This surrogate data can be created using the 2-min empty-room recordings. Beamforming coefficients can be used to estimate time series for each ROI based on the empty-room data and the same pipeline can be repeated. This would also be informative to know to what extent within-session fingerprinting accuracy is actually due to subject-independent signal components.

Individual traits: I suggest to perform fully cross-validated analysis where a model relating demographics and brain data is constructed on training data and then applied to test data. This

represents the current standard (and is also used in the work by Rosenberg cited by the authors (also used by many others)).

Figure 4 and 5: Please make brain plots larger (e.g. only three frequency bands per row)

Reviewer #2 (Remarks to the Author):

The present paper shows that the neurophysiological functional connectomes, as extracted by MEG recordings derived from a vast freely available dataset, enables the identification of subjects with very high identifiability rates, similar to those previously obtained using fMRI and EEG.

To the best of my knowledge, this is the first time that this very important question has been addressed using MEG analysis. The reported findings represent a relevant confirmation of what has been already observed using different neuroimaging modalities thus making this paper of big interest for a wide scientific community.

From a methodological point of view the paper is convincing and may drive a necessary shift towards the relevance to consider inter-subjects' variability in every clinical and basic neuroscience study. In this context, I have few minor remarks that the authors may consider in order to strengthen the conclusions.

Finally, the statistical analysis is valid, and, in my opinion, the authors made a great effort to make the reported finding fully reproducible.

Based on these comments, I would recommend accepting the present paper with minor revisions.

Minor remarks:

1- despite the possible variability induced by different approaches to (i) estimate functional connectivity, (ii) reconstruct source time-series or (iii) define an atlas has been discussed in previous works, it may be of interest to understand/discuss how these methodological choices may play a role in the subject-identification context. Furthermore, it would even be interesting to discuss how the use of different tasks (e.g., under naturalistic stimuli) may influence within- and between- subject similarity. I suppose most of these analyses are not possible to be addressed using this specific dataset but maybe it would still be relevant to discuss some of these points. Maybe, EEG studies, which for several reasons have deepened this research question in the context of biometric, may represent a valid starting point.

2- what do the authors think about a possible scenario where a specific subject or his/her fingerprint is not included in the matching set of data? How well would this approach allow to respond to this question? I mean, would it be possible to detect a sort of threshold in the feature-space to exclude his/her presence?

3- as for the healthy controls "only" performance, I think that this may represent a very important research question to be further addressed and probably, the reported findings even though derived from a very limited sample of "patients" can be deepened. In particular, it is licit to suppose that a specific brain disorder may affect neural fingerprint (making the identifiability more pronounced or less pronounced based on the specific disorder). Do the authors think would it be possible to report how the performance statistically differ between the two groups considering the class-balance problem?

Reviewer #3 (Remarks to the Author):

The paper by Castanheira et al reports results from a MEG fingerprinting study, in which the authors show that individuals can be recognised from their MEG data. Two separate methods of fingerprinting are employed - spectral (where the spectral content of MEG data across multiple parcellated regions are used) and connectome (where connectivity between all pairs of regions is computed using amplitude envelope correlation. Results show that fingerprinting is indeed possible using MEG data with >90% accuracy.

In this reviewer's opinion, this is an excellent, extremely exciting, and timely paper which should be published in Nature Communications. To my knowledge this is the first fingerprinting study attempted using MEG, and the results demonstrate high levels of performance using both spectral and connectome methods. In addition, the manuscript is extremely well written, the figures beautifully presented, and the methods sound. I have only two suggestions for the authors; the first is a methodological point which I would like to see addressed. The second more a philosophical point for the authors to ponder.

1) Leakage: On the connectome fingerprinting, my sincere apologies if I missed it but it looks like no steps were taken to remove leakage between the brain regions used. In my experience, connectome matrices can be dominated by leakage if it is not properly controlled and I worry that this might be affecting the results. In particular the authors conclude that it was connectivity between visual and limbic networks that best enabled fingerprinting. I worry however that this conclusion may be different if a leakage correction algorithm was used. I would suggest the symmetric orthogonalisation originally described by Colclough et al (NeuroImage 2015 117:439-48). However, pairwise orthogonalisation is also a useful approach (Hipp et al Nature Neuroscience, 2012) and should suffice.

2) In the last 10 years we have seen a paradigm shift in MEG (and neuroimaging in general). Whereas 'classical' MEG studies looked at task evoked changes and ignored ongoing activity – treating it as noise – we now recognise that this 'noise' has significant spatio-temporo-spectral structure which elucidates effects which are of significant behavioural consequence. Do we now stand at a similar "tipping point" with relation to between subject "noise"? A huge number of MEG studies scan many tens or perhaps hundreds of subjects from two or more cohorts (e.g. patients and controls). They then average together the results, and look for differences between groups. In other words, the between subject variance is treated as noise. However, the present study shows that this between subject variance contains interesting information (much like the ongoing activity that was also ignored for years!). Surely if this sensitivity can be harnessed we will see new clinical utility of neuroimaging emerge? E.g. rather than cross sectional, should we be looking to perform longitudinal studies in fewer patients to track changes – presumably if an individual can no longer be identified by their MEG fingerprint, that indicates a problem? So, I wondered if perhaps the authors might like to comment on this?

In sum, I am extremely keen to see this excellent paper published. I believe it to be novel, extremely interesting (not only to the MEG field but to scientists in general!) and I think it could become one of the cornerstones of a new paradigm in functional imaging.

Matt Brookes

RESPONSE TO REVIEWERS

We are grateful for the comments received from the Reviewers. We have brought substantial changes to address their concerns and followed their recommendations to the best of our abilities. In particular, we have performed a substantial number of new analyses and updated the majority of the figures. While the main body of results remain, we now provide what we think is further evidence of the novelty and potential impact of our approach and data. Below, we address each comment in detail and point at related changes in the revised manuscript.

Reviewer #1

1) Authors use resting-state MEG data from an open repository to show that identification of individuals based on their functional connectome or spectral signature is possible. This identification is performed on within-session data and between-session data. In addition, authors relate these fingerprints to demographic data. Overall, the research question is clearly described and relevant, the methodological approach is mostly appropriate and the results are largely convincing. Still, a number of concerns exist and are described in detail below.

Novelty is one concern. There is a sizeable literature of EEG-based biometric studies (even beyond refs 17,18). Some of them are based on resting-state data, use similar (or even more sophisticated) approaches and show similar identification accuracy. I am happy to acknowledge that the current study uses a large cohort and source-localised data but I suggest to discuss this study more explicitly in the context of the EEG literature. Authors state on page 12: 'The prospect of transferring MEG fingerprinting methodology to EEG as a more accessible sister technology is also appealing.' This is clearly incorrect since - if anything - the transfer happened in the opposite direction. Please emphasise the novel aspects of this study and acknowledge the existing EEG literature (particularly in the discussion).

We thank the Reviewer for their encouragement to better highlight related EEG literature and to better frame and explain the novelty of our present study with MEG.

Our study does advance brain fingerprinting in general and in particular fingerprinting based on electrophysiology data, in the several distinctive ways that follow.

First, as the Reviewer emphasizes, our study is based on a relatively large cohort of MEG data. A similar effort has yet to be published using EEG; we are aware of EEG fingerprinting studies with n=109 maximum, while our present study involved up to 158 participants (Fraschini et al., 2015; Kong et al., 2019; Rocca et al., 2014).

Second, we show that the proposed fingerprinting approach can identify individuals from data durations as short as 30 s, in what we refer to as the *short identification challenge*. Previous EEG attempts at fingerprinting have relied on recordings longer than 1-minute recordings (Fraschini et al., 2015; Kong et al., 2019; Rocca et al., 2014; fMRI studies were based on considerably longer durations for data acquisition). We also provide evidence that identification from such

short segments of data is reproducible days, weeks, or even years apart. We have reorganized the manuscript to better emphasize novelty in those regards. In addition, we elected to still report identification results from longer data segments for baseline validation purposes and better explained our rationale in the introduction:

“The within-session challenge with longer data segments was used to assess the baseline performances of the MEG fingerprinting approaches proposed. The more challenging situations investigated in the present report concern individual identification from shorter time segments of 30s within or between recording sessions.”

Third, previous EEG-fingerprinting work reproduced fMRI approaches based on connectome features. We are aware of only one EEG publication that used spectral-density features to derive fingerprinting, yet these features were used as an addition to the connectome approach (Rocca et al., 2014). Another of our original contributions is therefore to bring together both approaches (based on connectome vs. spectral measures) to assess their relative performances on a large, common dataset.

Fourth, we provide evidence that the fingerprinting approaches proposed in our manuscript are robust against recording artifacts, which are pervasive in electrophysiology and can bias identifiability. We believe this specific contribution has strong practical value and addresses a lingering concern from previous EEG-fingerprinting literature (Fraschini et al., 2015; Kong et al., 2019; Rocca et al., 2014).

And finally, we now provide empirical evidence of fingerprinting using scalp data vs. imaging models of their brain sources. A priori, these latter are more prone to modeling approximations and errors in EEG compared to MEG (Baillet, 2017; Leahy et al., 1998). Yet, our data show that MEG brain map models enable robust individual identification. Further, our imaging results based on cortical sources enable a more direct comparison with the fMRI-fingerprinting literature. These new results also clarify that environmental noise conditions that can change from session to session and therefore from participant to participant, do not contribute to identify individuals. Indeed, we believe our revised manuscript now shows more clearly how this latter aspect is addressed unambiguously with MEG, via same-day, empty-room recordings that have no EEG equivalent.

As mentioned above, we have substantially re-organized our manuscript to emphasize novelty and the other significant aspects listed above. Directly to the Reviewer’s comment, we have revised the Introduction and Discussion sections accordingly.

We have expanded on this point in the revised version of the Introduction:

“Previous EEG fingerprinting work was restricted to scalp data, and therefore, provided limited neuroanatomical insight (17–19). Another distinctive aspect of electrophysiology is the contamination of recordings by artefacts of different natures including environment and

instrument noise, muscle contractions, eye and head movements, which can be distinctive of individuals and can bias fingerprinting with non-neural signal features.”

We have also added to the revised Discussion:

“The methodological approaches proposed herein can, in principle, transfer to EEG fingerprinting (17–19), which would be more readily available in clinics. Whether results would be as robust with EEG than with MEG remains to be demonstrated. Indeed, EEG source mapping is more prone to contamination from muscle artifacts and is more sensitive to approximations in the biophysical modeling of head tissues, which may compromise further fingerprinting capabilities (27).”

2) Within-session versus between-session:

In general, I am not convinced by the within-session results.

First, afai the standard of this type of work in other modalities (e.g. fMRI) is always based on separate recordings. I don't see a good reason to change this here.

We agree with the Reviewer that between-session results such as those presented in our manuscript provide stronger (more robust) evidence of fingerprinting capabilities. Yet, we are also of the opinion that within-session data remain pertinent to the discussion of specific technical aspects related to the fingerprinting methodology. We agree with the Reviewer that the initial version of the manuscript did not make this distinction clear enough. We have therefore substantially revised the presentation of our within-session data to emphasize their value in the specific context of fingerprinting with electrophysiology/MEG. As a sidenote, there are several instances of fMRI fingerprinting studies that reported both within- and between-session data (e.g., Kaufmann et al., 2017; Miranda-Dominguez et al., 2014, 2018). The data in these papers were split into training and testing datasets based on volumes recorded on the same day, in addition to volumes recorded on separate days and used for identifying individuals. We also note that a substantial amount of the EEG fingerprinting literature is based on same-day recordings (e.g., Fraschini et al., 2015; Kong et al., 2019; Rocca et al., 2014). Somewhat related to the fingerprinting challenge, Rosenberg et al. (2016, 2020) used fMRI data collected from one single scanning session to identify individual's attention abilities using Connectome Predictive modeling (CPM).

We believe the significance and relevance of the within-session data and results reported in our manuscript are emphasized by the size of the cohort used for the identification challenges (n=158). Hence, we decided to still report the within-session results and better explained our reasoning in the introduction:

“The within-session challenge with longer data segments was considered to assess the baseline performances of the MEG fingerprinting approaches proposed. The more challenging situations developed in the present report concern individual identification from shorter 30-s time segments within or between recording sessions.”

In the present revised version, and following the Reviewer's suggestions, we also expanded substantially the analyses of within-session data and now provide a more detailed account of the putative contributions from environmental conditions to fingerprinting performances, as explained and detailed further below. We also clarified and justified in the main body of the manuscript the rationale for featuring the within-session data.

3) Second, it is difficult to ensure that within-session results are not caused or contaminated by effects that are independent of the individual. Imagine some external noise source on the day of measurement that is picked up by the MEG system and introduces a spectral structure (or connectivity structure) that leads to high correlation for the within-session fingerprinting. In fact, figure 2c shows that performance degrades with increasing temporal distance between the short data blocks. This suggests that some slow-changing features exist in the data that apparently boost within-session correlation. I currently don't see how a strong case for fingerprinting can be made based on within-session correlations.

In summary, I suggest to remove the within-session results or at least move them to Suppl Mat. If authors disagree then I would expect to see a good argument for this approach (which is apparently not the standard (e.g. in fMRI)).

We truly appreciate and agree with the point raised by this Reviewer, as we already alluded to in the replies above. We therefore performed new analyses that confirmed that the identification effects reported are robust against day-specific, environmental noise conditions. These new results are integrated and discussed in the revised version of the manuscript and explained in detail below.

First, regarding slowly evolving environmental noise conditions potentially increasing identifiability from within-session data artificially, we argue there would be a contradiction in these latter both increasing within-session identifiability from longer data segments and decreasing identifiability from shorter data segments extracted from the same session. Indeed, the shorter segments used for identification were extracted from the longer within-session data used for the *within-session* challenge. For environmental noise conditions to drive within-session identifiability, they would need to remain virtually identical over the duration of the recording session but would need to differ substantially across sessions/individuals. Hence in these putative conditions and for each individual, environmental noise would remain stationary across each of the 30-s segments used for the shortened within-session challenge. Therefore, we would not expect to observe decreased identification performances from shorter vs. longer segments from the same session, although this is observed in our data. Further, our results indicate that identification from faster frequency components performs better than slower frequency bands (e.g., delta). We argue this is also in contradiction with the eventuality of slow environmental, not brain, fluctuations contributing to individual identification.

Second, we performed new analyses for this revision to demonstrate that individuals cannot be identified from the environmental conditions on the day of their respective sessions. As the Reviewer kindly suggested (see also Point #7 below from this Reviewer), we processed empty-

room recordings in an identical manner as the MEG data collected in presence of participants. We applied the same individual beamformer weights to empty-room sensor data as those used for imaging actual brain recordings. These new data show that individuals cannot be identified from empty-room recordings of environmental conditions alone (overall performance <20% at the source level; revised Figure 2). In particular, and directly to the Reviewer's point, the data also emphasize that there are no slow-changing features in the empty-room recordings that would boost within-session identification.

These results are discussed in the Results and Supplemental Material, respectively:

"We first processed each individual session's empty-room recordings in an identical fashion to participants brain data. In particular, we produced pseudo brain maps of empty-room sensor data using the same imaging kernels as those used for each session's participant brain data. The implication is that imaging kernels designed based on information that are specific of each participant, such as their respective head positions in the MEG sensor array and individual anatomy brain features that constrain MEG source maps. We therefore tested whether such individual information unrelated to brain activity contributed substantially to individual identification from MEG source maps. We found that identification performances were considerably reduced using empty-room data (<20% across all tested models; Figure 2). These results based on source maps were corroborated by the low fingerprinting performances obtained by using empty-room sensor data only (<5% across all tested models; Supplemental Material)."

*"Individuals cannot be identified from their respective imaging kernels
We verified that the within-session identification of individuals was not possible from empty-room data (i.e., with no participant under the MEG sensor array) processed through their respective imaging kernel of beamformer weights. Indeed, these latter are defined from individual anatomy and head position under the MEG sensor array, which may have been sufficient information to drive identification. We therefore ran the same fingerprinting pipeline on each session's empty-room data transformed through the corresponding individual's beamformer imaging kernel, which was identical for each of the within-session data segments used. Note that for the between-session challenges, the imaging kernels were adjusted to the respective individual head positions measured during each session. These analyses demonstrated that the imaging kernel information did not contribute substantially to MEG fingerprinting (overall performance was below 20% on average)."*

For these additional reasons, we believe that the within-session results contribute to showcasing MEG's fingerprinting abilities in a large cohort of individuals, and that these abilities are not biased by environmental conditions or source imaging kernel models. We believe these new results are also strengthened by new identification analyses directly from sensor data i.e., with no imaging kernel, which are detailed in the revised Supplemental Material. The sensor-level within-session challenge shows robust identification performances (overall >80%). Although performances are decreased for the sensor-level between-session challenge (yet they overall remained >50%) possibly due to changes in head position between sessions, sensor level

fingerprinting performs well above the empty-room conditions as described above (source level <20%). In addition, and as matter of further validation of the approach, we performed empty-room sensor fingerprinting by attempting to identify individuals from sensor empty-room data, which produced very low identification rates of <5% (see Supplemental Figure 8).

We discuss these results in detail in the Supplemental Material:

“We also ran the MEG fingerprinting pipeline directly from the sensor data of the empty-room recordings, without transformation through individual imaging kernels, to assess the floor level of identification performances from non-brain data only. The data confirmed substantially lower levels of identification (<5% accuracy on average; see Figure S8).

Fingerprinting from scalp data only

We also performed MEG fingerprinting from individual sensor data, with no MEG source reconstruction to assess the added value of source modeling. We replicated the above MEG fingerprinting pipelines from the within-, within-shortened, and between- session analyses. Identification performances were less than with source modeling, especially from signal components in higher frequency bands and for the shortened challenges (see Figure S9, S10, & S11). Yet for other signal components and longer durations, individuals remain identifiable from sensor-level data collected between sessions (>60% accuracy from broadband data), albeit with lower accuracy than when using MEG source transformations, which explicitly account for different head positions between sessions.

Taken together with the empty-room fingerprinting tests above, these results provide evidence that brain signals, not environmental conditions, were crucial for individual identification.”

For all these reasons, we believe these new results provide strong evidence that the identification performances obtained from the within-session data challenge i) are pertinent and informative and ii) are due to genuine brain activity, not fluctuations of external noise contributions.

4) Related to the previous point: If authors want to make the claim that 30s are sufficient for fingerprinting then this should be shown for between-session data as well.

We thank the Reviewer for this suggestion and agree with the recommendation. We therefore performed new between-session identification analyses based on shortened data segments and moved the previous within-session shortened analysis to the Supplemental Material. We used 30-s data from the first and second sessions whenever available from individuals in the cohort. Revised Figure 3 shows the new results, which corroborate the robustness of MEG's fingerprinting ability from short data segments.

Based on these and the above new analyses and verifications, we do believe the claim can be made that fingerprinting can be obtained from short 30-s MEG data segments obtained longitudinally over distinct recording sessions.

5) All results are based on the correlation matrix of feature vectors within and between participants. Please show an example matrix in Suppl Mat.

Thank you for this suggestion. We now provide a typical instance of a correlation matrix used for fingerprinting in Supplemental Material. The array plot shows how participants are correlated to themselves (over the diagonal) and between each other (off diagonal elements) for the neural features tested.

6) Please add methodological details: What type of beamformer was used? Any regularisation? How was the empty-room recording used? For the Welch PSD provide frequency resolution and frequency range.

We have augmented the Methods section to provide the methodological details requested. In short, we used LCVM beamformers regularized using linear scaling of data covariance, all based on the default parameters of the Brainstorm general distribution. The beamformers output was normalized using the noise covariance matrix to derive the pseudo neural activity index, as recommended by Liu and Van (1992) and Van Veen et al. (1997). The implementation details in Brainstorm are identical to those provided in these original publications, as detailed the online documentation.

Power Spectrum Density estimates were derived using Welch's method, also using Brainstorm's default implementation. The frequency range was from 0 to 150 Hz, with a frequency resolution of 0.5 Hz.

7) It would be very helpful to estimate the expected accuracy in the absence of any real correlations between data1 and data2 due to subject-specific brain activity. This surrogate data can be created using the 2-min empty-room recordings. Beamforming coefficients can be used to estimate time series for each ROI based on the empty-room data and the same pipeline can be repeated. This would also be informative to know to what extent within-session fingerprinting accuracy is actually due to subject-independent signal components.

We thank the Reviewer for their suggestion of using empty-room recordings as surrogate data. As briefly mentioned above (see Point #3), we performed a new set of fingerprinting analyses based on sensor data only, hence not augmented by a source modeling procedure. As mentioned above, fingerprinting performances were high (overall identification accuracy >80%) in the *within-session* challenge and remained strong based on *between-session* data (overall identification accuracy >50%).

We also performed new individual fingerprinting analyses based on empty-room environment recordings. As explained above (see Point #3), the new data demonstrate that individuals cannot be identified from session-specific environmental conditions within- or between- sessions (revised Figure 2 and Supplemental Material), even when empty-room sensor data is augmented by beamforming. The overall empty-room fingerprinting performances at the source level ranged below 20% across all frequency bands tested, which provide informative lower threshold

values of identifiability that we now report in all relevant figures featuring individual fingerprinting results.

We can therefore conclude that within-session results are specific of individual brain data and are not driven by environmental conditions. Note also that the imaging kernels (beamformer weights) were adjusted with each session's empirical statistics of recorded noise and brain data, as explained in Point #6.

8) Individual traits: I suggest to perform fully cross-validated analysis where a model relating demographics and brain data is constructed on training data and then applied to test data. This represents the current standard (and is also used in the work by Rosenberg cited by the authors (also used by many others)).

We thank the Reviewer for this suggestion. We agree that cross-validation of our PLS analysis would provide stronger evidence that these effects are generalizable out of sample. Rosenberg et al. (2016) performed CPM to predict outcomes, and while our approach has similarities their analysis, the respective techniques and rationales are slightly different. First and foremost, the CPM analyses require cross-validation to make sure the predictions generalize outside of the sample. Our goal was to demonstrate whether some of the fingerprinting features used in our data may have been related to demographics. The purpose was to investigate with MEG, as reported elsewhere with other imaging modalities, that individual differences in connectome and spectral features may relate to individual differences in traits and abilities (e.g., Clark et al., 2004; Rosenberg et al., 2016, 2017, 2020). As mentioned in the manuscript, our PLS analysis was therefore exploratory and was not intended to test a specific directional hypothesis concerning demographics. To the point concerning generalizability or lack thereof, we kindly remind the Reviewer that we had performed permutation tests and bootstrapping to assess significance of latent variables and robustness of the latent factors.

To thoroughly cross-validate our current results from the PLS exploratory analysis, we foresee that a distinct dataset such as CAMCAN or any other large MEG repository should be used in future work.

9) Figure 4 and 5: Please make brain plots larger (e.g. only three frequency bands per row)

We have revised the figure per the Reviewer's suggestion.

Reviewer #2 (Remarks to the Author):

The present paper shows that the neurophysiological functional connectomes, as extracted by MEG recordings derived from a vast freely available dataset, enables the identification of subjects with very high identifiability rates, similar to those previously obtained using fMRI and EEG. To the best of my knowledge, this is the first time that this very important question has been addressed using MEG analysis. The reported findings represent a relevant confirmation of what has been already observed using different neuroimaging modalities thus making this paper of big

interest for a wide scientific community.

From a methodological point of view the paper is convincing and may drive a necessary shift towards the relevance to consider inter-subjects' variability in every clinical and basic neuroscience study. In this context, I have few minor remarks that the authors may consider in order to strengthen the conclusions.

Finally, the statistical analysis is valid, and, in my opinion, the authors made a great effort to make the reported finding fully reproducible.

Based on these comments, I would recommend accepting the present paper with minor revisions.

Minor remarks:

1- despite the possible variability induced by different approaches to (i) estimate functional connectivity, (ii) reconstruct source time-series or (iii) define an atlas has been discussed in previous works, it may be of interest to understand/discuss how these methodological choices may play a role in the subject-identification context. Furthermore, it would even be interesting to discuss how the use of different tasks (e.g., under naturalistic stimuli) may influence within and between- subject similarity. I suppose most of these analyses are not possible to be addressed using this specific dataset but maybe it would still be relevant to discuss some of these points. Maybe, EEG studies, which for several reasons have deepened this research question in the context of biometric, may represent a valid starting point.

We agree with the Reviewer that these are important questions to investigate in future studies. As mentioned in their comment above, we believe these are questions that reach beyond the scope of the present paper as they would require additional and substantial new data collection. We do believe however that our present results may inspire and pave the way for further studies in that direction.

We have revised and augmented the Discussion in that respect (see below). Specifically, we now cite a recent preprint related to these questions, and which specifically discusses the choice of functional connectivity measures for individual fingerprinting e.g., based on amplitude vs. phase measures across frequency bands (Sareen et al., 2021).

We have expanded on this point in the revised version of the Discussion:

“Importantly, our present work does not address how certain methodological choices may affect our ability to fingerprint. While we demonstrate the robustness of our current approach, in addition to demonstrating that the method used for source imaging (i.e., the beamformer imaging kernel) cannot identify individuals and that sensor level fingerprinting remains successful, it remains unclear how different source estimation approaches may alter our results. In addition, choices with regards to connectivity measures may also play a role in influencing identifiability. Current findings have indeed begun to explore how connectivity measures may influence ones' fingerprinting capacities (48).”

In the present revised version however, we now provide new results derived from MEG sensor data only i.e., with no beamforming/source imaging treatment, which show that individuals can still reliably be identified across sessions (see Supplemental Material). In addition, we also assessed if the imaging kernel used for source imaging in the *within-session* challenge may be sufficient to identify individuals (see Point #7 from Reviewer-1 above). To do so we used the empty-room recordings collected around each individual session and produced their corresponding source maps akin to how we processed actual brain data. The revised manuscript shows that individuals cannot be identified from the resulting source maps e.g., empty-room recordings processed via source imaging beamformers. Taken together, we believe these new results demonstrate the identification of individuals with minimal treatment of the simplest form of MEG brain data (i.e., sensor data), and hence substantial robustness against modeling choices involved in e.g., source modeling.

We have expanded on this point in the revised version of the Results:

“We investigated the robustness of these results against variables of no interest and possible confounds. We first processed each individual session’s empty-room recordings in an identical fashion to participants brain data. In particular, we produced pseudo brain maps of empty-room sensor data using the same imaging kernels as those used for each session’s participant brain data. The implication is that imaging kernels designed based on information that are specific of each participant, such as their respective head positions in the MEG sensor array and individual anatomy brain features that constrain MEG source maps. We therefore tested whether such individual information unrelated to brain activity contributed substantially to individual identification from MEG source maps. We found that identification performances were considerably reduced using empty-room data (<20% across all tested models; Figure 2). These results based on source maps were corroborated by the low fingerprinting performances obtained by using empty-room sensor data only (<5% across all tested models; Supplemental Material).”

We have also expanded on this point in the revised version of the Discussion:

“The data also indicates that MEG fingerprinting is robust against typical recording artefacts that may be idiosyncratic of individuals and therefore, could have confounded identification. In particular, session environmental conditions captured by empty-room MEG recordings were not sufficient to identify individuals within or between sessions. The participant’s anatomical and head-position information embedded in their respective MEG source imaging kernels were also not sufficient to identify individuals. Note that head position changed between sessions. Further studies are required to clarify how these results may vary depending on the type of MEG source modelling adopted. We anticipate little influence of the type of source model used though, based on evidence that beamforming kernels are mathematically equivalent to other major classes of linear source estimation kernels, such as weighted minimum-norm estimators (58). Future work should corroborate these results with regards to fingerprinting. The choice of connectivity measure to derive electrophysiological connectomes may also influence identifiability (59). We look forward to current progress in electrophysiological brain connectomics to put forward

measures of network connectivity informed by mechanistic principles and emerging as a standard metrics in the field to confirm and expand present fingerprinting results (60). ”

2- what do the authors think about a possible scenario where a specific subject or his/her fingerprint is not included in the matching set of data? How well would this approach allow to respond to this question? I mean, would it be possible to detect a sort of threshold in the feature-space to exclude his/her presence?

We thank the Reviewer for this intriguing question concerning the identification of a possible “outlier”. However, we believe our fingerprinting approach, which is based on the highest correlation score between features, is not well suited to address this question. Indeed, the

identification procedure would de facto assign the identity of the *outlier* person to that of a participant in the cohort with features that are the most correlated to the outlier. One possibility would be to establish a threshold of minimum inter-individual correlation to establish identifiability. However, the choice of this threshold’s value would be arbitrary and may vary depending on the cohort of individuals to be identified (e.g., between siblings or relatives, per age group, clinical population, etc.).

Overall, our study’s goal was not to derive a brain fingerprinting approach for purposes of forensic identification, as in a biometric system. This may be the objective of subsequent studies conducted by inspired readers. Instead, our data provide evidence that different electrophysiological features are sufficiently stable within the same individual across (possibly long periods of) time to make them identifiable. We foresee that the present study will encourage future work for testing the stability or evolution of individual brain fingerprinting features over time e.g., in response to treatment, during growth, development, learning and aging.

3- as for the healthy controls “only” performance, I think that this may represent a very important research question to be further addressed and probably, the reported findings even though derived from a very limited sample of “patients” can be deepened. In particular, it is licit to suppose that a specific brain disorder may affect neural fingerprint (making the identifiability more pronounced or less pronounced based on the specific disorder). Do the authors think would it be possible to report how the performance statistically differ between the two groups considering the class-balance problem?

We subscribe to the Reviewer’s vision. As mentioned in Point #2 above, we plan on exploring how the present fingerprinting approach may transfer to the evaluation of clinical disorders. For instance, previous work indicates that mental disorders make neural fingerprinting less stable across time (Kaufmann et al., 2017, 2018). The cohort used in the present seminal study did not yield substantial distinction between subgroups yet, but that was not the primary purpose of our investigations.

More specifically, our data feature an unequal number of participants per class (i.e., healthy controls vs. patients) with two sub-classes of patient subpopulations, and therefore make the assessment of fingerprinting classification accuracy between groups particularly challenging or ill-designed. We believe that bootstrapping individuals from the healthy control group to match the number of patients would not be fair because the healthy controls featured in our cohort were more heterogenous in age, among other factors, than the patients in the cohort available to us.

We anticipate that future specific studies of appropriate datasets, matched in size and general demographics between patients and controls, will fully appreciate the value of brain fingerprinting approaches in clinically-oriented research.

We have expanded on this point in the revised version of the Discussion:

“Our present study highlights the utility of datasets with highly samples individuals to harness the potential of between participant variance and demonstrates that what was once considered ‘subject noise’ to be averaged out may in fact be informative. Ideally, large databanks of individual variants sampled across multiple dimensions of socio-economic, age, and geographic factors enable normative modeling approaches to establish the risk traits of developing syndromes of e.g., early cognitive decline, neurodegeneration, or mental illness. Indeed, previous work has indicated that certain mental illnesses may in fact render individuals less identifiable over time (15, 68). Stability in an individual’s unique neurophysiological features may represent a useful clinical marker for future work.”

Reviewer #3 (Remarks to the Author):

The paper by Castanheira et al reports results from a MEG fingerprinting study, in which the authors show that individuals can be recognised from their MEG data. Two separate methods of fingerprinting are employed - spectral (where the spectral content of MEG data across multiple parcellated regions are used) and connectome (where connectivity between all pairs of regions is computed using amplitude envelope correlation). Results show that fingerprinting is indeed possible using MEG data with >90% accuracy.

In this reviewers opinion, this is an excellent, extremely exciting, and timely paper which should be published in Nature Communications. To my knowledge this is the first fingerprinting study attempted using MEG, and the results demonstrate high levels of performance using both spectral and connectome methods. In addition, the manuscript is extremely well written, the figures beautifully presented, and the methods sound. I have only two suggestions for the authors; the first is a methodological point which I would like to see addressed. The second more a philosophical point for the authors to ponder.

1) Leakage: On the connectome fingerprinting, my sincere apologies if I missed it but it looks like no steps were taken to remove leakage between the brain regions used. In my experience, connectome matrices can be dominated by leakage if it is not properly controlled and I worry that

this might be affecting the results. In particular the authors conclude that it was connectivity between visual and limbic networks that best enabled fingerprinting. I worry however that this conclusion may be different if a leakage correction algorithm was used. I would suggest the symmetric orthogonalisation originally described by Colclough et al (NeuroImage 2015 117:439-48). However, pairwise orthogonalisation is also a useful approach (Hipp et al Nature Neuroscience, 2012) and should suffice.

We thank the Reviewer for bringing up this valid technical point. We acknowledge that spatial leakage affects electromagnetic source reconstruction from sensor data and may confound the anatomical interpretation of MEG source maps from sensor data. However, we elected not to orthogonalize source time series prior to computing AEC measures for the following reasons.

First, we wished to maximize consistency between the time series used for the connectome and spectral fingerprinting approaches. We therefore opted for not orthogonalizing signals prior to deriving AEC connectivity statistics, such that the connectivity fingerprinting pipeline would be as similar as possible to that of spectral fingerprinting for which we did not orthogonalize signals prior to estimating power spectrum densities.

Second, we emphasize that the beamforming approach we used for source mapping is designed to minimize spatial leakage between regions where activity would be strongly correlated with zero-lag delays (Van Veen et al., 1997). In essence, its purpose is therefore similar to time series orthogonalization.

Third, we aimed to use a straightforward connectivity measure akin to the correlational techniques applied in fMRI fingerprinting. While there may be some concern that different connectivity measures can yield different results, we discuss the possibility for future work to address this concern.

We expand on this point in the revised version of the Discussion:

“The data also indicates that MEG fingerprinting is robust against typical recording artefacts that may be idiosyncratic of individuals and therefore, could have confounded identification. In particular, session environmental conditions captured by empty-room MEG recordings were not sufficient to identify individuals within or between sessions. The participant’s anatomical and head-position information embedded in their respective MEG source imaging kernels were also not sufficient to identify individuals. Note that head position changed between sessions. Further studies are required to clarify how these results may vary depending on the type of MEG source modelling adopted. We anticipate little influence of the type of source model used though, based on evidence that beamforming kernels are mathematically equivalently to other major classes of linear source estimation kernels, such as weighted minimum-norm estimators (58). Future work should corroborate these results with regards to fingerprinting. The choice of connectivity measure to derive electrophysiological connectomes may also influence identifiability (59). We look forward to current progress in electrophysiological brain connectomics to put forward measures of network connectivity informed by mechanistic principles and emerging as a standard metrics in the field to confirm and expand present fingerprinting results (60). ”

We acknowledge that spatial leakage tends to blur individual brain maps, making them more similar across individuals. Intuitively, this would make fingerprinting more challenging, although our data show this challenge is tractable. We also note that recent reports have shown that signal orthogonalization may negatively affect fingerprinting performances. Sareen et al. (2021) have indeed shown that uncorrected (non-orthogonalized) AEC measures yield better identification. We also note that the same study used an ICC analysis similar to our present study, and emphasized that occipital visual regions were also key to fingerprinting performances. Another study also showed that uncorrected AEC outperformed corrected AEC in identifying pairs of monozygotic twins (Demuru et al., 2017). Based on these and the considerations above, we elected not to apply time series orthogonalization corrections.

2) In the last 10 years we have seen a paradigm shift in MEG (and neuroimaging in general). Whereas 'classical' MEG studies looked at task evoked changes and ignored ongoing activity – treating it as noise – we now recognise that this 'noise' has significant spatio-temporo-spectral structure which elucidates effects which are of significant behavioural consequence. Do we now stand at a similar "tipping point" with relation to between subject "noise"? A huge number of MEG studies scan many tens or perhaps hundreds of subjects from two or more cohorts (e.g. patients and controls). They then average together the results, and look for differences between groups. In other words, the between subject variance is treated as noise. However, the present study shows that this between subject variance contains interesting information (much like the ongoing activity that was also ignored for years!). Surely if this sensitivity can be harnessed we will see new clinical utility of neuroimaging emerge? E.g. rather than cross sectional, should we be looking to perform longitudinal studies in fewer patients to track changes – presumably if an individual can no longer be identified by their MEG fingerprint, that indicates a problem? So, I wondered if perhaps the authors might like to comment on this?

We thank the Reviewer for this insightful comment and perspective. Indeed, MEG studies, akin to other modalities of neuroimaging and in neuropsychology in general, have typically considered inter-participant variability as noise or a form of nuisance of no interest. We believe our present study contributes to reconsidering inter-individual variance as a meaningful source of information. As the Reviewer emphasizes, we also hope our present contribution encourages future studies with a similar perspective on MEG/EEG research. As the Reviewer also points out, recent fMRI findings suggest that individuals may become less identifiable over time due to the development of mental disorders (Kaufmann et al., 2017, 2018). Thus, the ability to brain-fingerprint individuals using electrophysiological features that are expected to be stable over time may translate into a family of practical clinical signal markers. For the time being, these aspects remain speculative and require further research with the adequate data cohorts.

We have revised the Discussion to further elaborate on these significant aspects:

"Our present study highlights the utility of datasets with highly samples individuals to harness the potential of between participant variance and demonstrates that what was once considered 'subject noise' to be averaged out may in fact be informative. Ideally, large databanks of individual variants sampled across multiple dimensions of socio-economic, age, and geographic

factors enable normative modeling approaches to establish the risk traits of developing syndromes of e.g., early cognitive decline, neurodegeneration, or mental illness. Indeed, previous work has indicated that certain mental illnesses may in fact render individuals less identifiable over time (15, 68). Stability in an individual's unique neurophysiological features may represent a useful clinical marker for future work."

In sum, I am extremely keen to see this excellent paper published. I believe it to be novel, extremely interesting (not only to the MEG field but to scientists in general!) and I think it could become one of the cornerstones of a new paradigm in functional imaging.

References

- Baillet, S. (2017). Magnetoencephalography for brain electrophysiology and imaging. *Nature Neuroscience*, 20(3), 327–339. <https://doi.org/10.1038/nn.4504>
- Demuru, M., Gouw, A. A., Hillebrand, A., Stam, C. J., van Dijk, B. W., Scheltens, P., Tijms, B. M., Konijnenberg, E., ten Kate, M., den Braber, A., Smit, D. J. A., Boomsma, D. I., & Visser, P. J. (2017). Functional and effective whole brain connectivity using magnetoencephalography to identify monozygotic twin pairs. *Scientific Reports*, 7(1), 9685. <https://doi.org/10.1038/s41598-017-10235-y>
- Fraschini, M., Hillebrand, A., Demuru, M., Didaci, L., & Marcialis, G. L. (2015). An EEG-Based Biometric System Using Eigenvector Centrality in Resting State Brain Networks. *IEEE Signal Processing Letters*, 22(6), 666–670. <https://doi.org/10.1109/LSP.2014.2367091>
- Kaufmann, T., Aln s, D., Brandt, C. L., Bettella, F., Djurovic, S., Andreassen, O. A., & Westlye, L. T. (2018). Stability of the Brain Functional Connectome Fingerprint in Individuals With Schizophrenia. *JAMA Psychiatry*, 75(7), 749. <https://doi.org/10.1001/jamapsychiatry.2018.0844>
- Kaufmann, T., Aln s, D., Doan, N. T., Brandt, C. L., Andreassen, O. A., & Westlye, L. T. (2017). Delayed stabilization and individualization in connectome development are related to psychiatric disorders. *Nature Neuroscience*, 20(4), 513–515. <https://doi.org/10.1038/nn.4511>
- Kong, W., Wang, L., Xu, S., Babiloni, F., & Chen, H. (2019). EEG Fingerprints: Phase Synchronization of EEG Signals as Biomarker for Subject Identification. *IEEE Access*, 7, 121165–121173. <https://doi.org/10.1109/ACCESS.2019.2931624>
- Leahy, R. M., Mosher, J. C., Spencer, M. E., Huang, M. X., & Lewine, J. D. (1998). A study of dipole localization accuracy for MEG and EEG using a human skull phantom. *Electroencephalography and Clinical Neurophysiology*, 107(2), 159–173. [https://doi.org/10.1016/S0013-4694\(98\)00057-1](https://doi.org/10.1016/S0013-4694(98)00057-1)

Miranda-Dominguez, O., Feczko, E., Grayson, D. S., Walum, H., Nigg, J. T., & Fair, D. A. (2018). Heritability of the human connectome: A connectotyping study. *Network Neuroscience* (Cambridge, Mass.), 2(2), 175–199. https://doi.org/10.1162/netn_a_00029

Miranda-Dominguez, O., Mills, B. D., Carpenter, S. D., Grant, K. A., Kroenke, C. D., Nigg, J. T., & Fair, D. A. (2014). Connectotyping: Model based fingerprinting of the functional connectome. *PloS One*, 9(11), e111048. <https://doi.org/10.1371/journal.pone.0111048>

Richard Clark, C., Veltmeyer, M. D., Hamilton, R. J., Simms, E., Paul, R., Hermens, D., & Gordon, E. (2004). Spontaneous alpha peak frequency predicts working memory performance across the age span. *International Journal of Psychophysiology*, 53(1), 1–9. <https://doi.org/10.1016/j.ijpsycho.2003.12.011>

Rocca, D. L., Campisi, P., Vegso, B., Cserti, P., Kozmann, G., Babiloni, F., & Fallani, F. D. V. (2014). Human Brain Distinctiveness Based on EEG Spectral Coherence Connectivity. *IEEE Transactions on Biomedical Engineering*, 61(9), 2406–2412. <https://doi.org/10.1109/TBME.2014.2317881>

Rosenberg, M. D., Finn, E. S., Scheinost, D., Constable, R. T., & Chun, M. M. (2017). Characterizing Attention with Predictive Network Models. *Trends in Cognitive Sciences*, 21(4), 290–302. <https://doi.org/10.1016/j.tics.2017.01.011>

Rosenberg, M. D., Finn, E. S., Scheinost, D., Papademetris, X., Shen, X., Constable, R. T., & Chun, M. M. (2016). A neuromarker of sustained attention from whole-brain functional connectivity. *Nature Neuroscience*, 19(1), 165–171. <https://doi.org/10.1038/nn.4179>

Rosenberg, M. D., Scheinost, D., Greene, A. S., Avery, E. W., Kwon, Y. H., Finn, E. S., Ramani, R., Qiu, M., Constable, R. T., & Chun, M. M. (2020). Functional connectivity predicts changes in attention observed across minutes, days, and months. *Proceedings of the National Academy of Sciences of the United States of America*, 117(7), 3797–3807. <https://doi.org/10.1073/pnas.1912226117>

Sareen, E., Zahar, S., Van De Ville, D., Gupta, A., Griffa, A., & Amico, E. (2021). Exploring MEG brain fingerprints: Evaluation, pitfalls, and interpretations [Preprint]. *Neuroscience*. <https://doi.org/10.1101/2021.02.15.431253>

Van Veen, B. D., Van Drongelen, W., Yuchtman, M., & Suzuki, A. (1997). Localization of brain electrical activity via linearly constrained minimum variance spatial filtering. *IEEE Transactions on Biomedical Engineering*, 44(9), 867–880. <https://doi.org/10.1109/10.623056>

REVIEWER COMMENTS

Reviewer #1 (Remarks to the Author):

Authors have convincingly addressed my comments and I am happy to recommend publication.

Reviewer #2 (Remarks to the Author):

In my opinion the authors did a great job and have addressed all the previous remarks. I have no further comments.

Matteo Fraschini

Reviewer #3 (Remarks to the Author):

I thank the authors for addressing my comments.

I remain somewhat concerned about signal leakage. However, I completely understand, and respect, the authors reasoning for not applying orthogonalisation. Ultimately it is their decision and their motivation has been made clear.

Once again my congratulations to the authors on an excellent paper which I think will drive our research field forward considerably.